# Effects of Different Infra-Red Irradiations on the Survival of Granary Weevil *Sitophilus granarius*: Bioefficacy and Sustainability

**DOI:** 10.3390/insects12020102

**Published:** 2021-01-25

**Authors:** Sándor Keszthelyi, Helga Lukács, Ferenc Pál-Fám

**Affiliations:** Department of Plant Production and Protection, Institute of Plant Science, Faculty of Agricultural and Environmental Sciences, Kaposvár University, H-7400 Kaposvár, Hungary; lukacs.helga@szie.hu (H.L.); pff3pff3@gmail.com (F.P.-F.)

**Keywords:** activity changing, environmentally-friendly pest management, granary weevil, IR, mortality, progeny, stored product pest

## Abstract

**Simple Summary:**

The granary weevil *Sitophilus granarius* is a cosmopolitan insect that causes substantial damage to many stored products. Control methods based on the use of sustainable, residual insecticide-free methods are gaining importance due to their features such as bioactivity, biodegradability, and ecological safety. However, despite the finding that different irradiation methods have proven efficacious, infrared treatment has not gained widespread application, as different aspects of its application have not yet been explored. In this study, we evaluated the activity-inducing and mortality effect profile of infrared irradiation on *S. granarius*. The impact of the distance between the radiation emitter and the treated material on the mortality rate was tested in search of optimal results. Adult-perishing and progeny-suppressive effects caused by infrared irradiation in *S. granarius* were confirmed. Our results point out that the efficacy of treatment depends on the optimal distance between the treated specimen and the emitter. Information on adult-activity triggered by radiation facilitates the application of other control methods and the optimization of their timing. In summary, our findings indicate that the use of infrared irradiation for the post-harvest protection of cereals is feasible, which can contribute to the realization of endeavors aimed at environmentally friendly and residuum-free pest management.

**Abstract:**

*Sitophilus granarius* (L.) is an important pest of stored grain worldwide. In recent years, sustainable methods against it have received attention as grain stock protective means. Our aim was to obtain information about the efficacy of infrared irradiation (IR) against *S. granarius* in laboratory conditions. The change in adult-activity and median lethal dose (LD_50_) triggered by IR in *S. granarius* was examined. The insecticidal efficacy in the infested grains was also analyzed at 12, 24, 48, and 72h following exposure to IR (250W), and the progeny-production was assessed 45 days upon the treatment. Based on our findings, total mortality ensued in a grain stock of 50 g at 412 s and a for 100 g grain at 256 s. A significant increase in *S. granarius* mortality could be observed in the higher grain weight regime, which can be accounted for by the higher heat-absorbance of objects with higher weight. The activity of pests immediately after the beginning of IR increased and subsequently became moderated. The observation of activity-peak brought about by irradiation contribute to the optimization of chemical intervention. This treatment could provide an effective and sustainable technique in integrated pest management.

## 1. Introduction

The granary weevil *Sitophilus granarius* L. (Coleoptera: Curculionidae) is a typical cosmopolitan pest. It can cause significant damage to stored grains and can drastically decrease yields [1]. Damaged grain has reduced nutritional and market value, low percentage germination, and reduced weight [2]. Its importance is also emphasized by its primary pest role because it can open a way for other secondary pests [1].

Regrettably, there are several side effects in the application of residual insecticides used by pest management in stored products. They can be toxic to mammals, residues can accumulate in treated products, and many pest species could become resistant against protectants [3]. Recently, sustainable methods have received increased attention as grain protective possibilities [4].

As the regular practice (utilization of residual insecticides) becomes more and more limited, irradiation is likely to gain increasing attention as an alternative way of plant protection in the future [5]. Therefore, stored product-pest sterility and mortality induced by various types of electromagnetic irradiation such as ionizing, microwave, or infrared radiation can be considered [6].

Infrared irradiation (IR) is electromagnetic radiation (EMR) with wavelengths longer than those of visible light (800–4500 nm). It is heat radiation and subsequently will heat surfaces that absorb them [6,7]. Heat is a sort of energy transmission that flows due to a temperature difference. Thermal radiation is characterized by a particular spectrum of many wavelengths that are associated with emission from an object due to the vibration of its molecules at a given temperature [8]. Infrared radiation is used widely, e.g., in thermography, spectroscopy, climatology, meteorology, and astronomy. Its special effects can be applied in heating as it predominantly heats the opaque, absorbent objects rather than the air around them [9,10,11]. 

The benefits of infrared technology can be utilized in plant protection practices in several ways, such as in the detection of insect infestation in stored products [12,13,14]. Another way is the determination of quality change in food caused by insects [15] or in direct pest management. Numerous efforts have already been made in its practical application in connection with its utilization to combat several stored product pests, for instance, *Sitotroga cerealella* (Olivier, 1789) [16], *Rhyzoperta dominica*, Fabricius, 1792 [17], *Sitophilus oryzae* (L., 1763) [18], *Sitophilus zeamais* (Motschulsky, 1855) and *Tribolium castaneum* (Herbst, 1797) [19].

Nevertheless, relevant information about *S. granarius* in this regard is rather scarce. Therefore, the objectives of this research were to obtain information about the efficacy of different exposure times and the median lethal doses of infrared irradiation against *S. granarius* on different masses of wheat grain, and to evaluate the influence of different times elapsed after treatment of applied doses on weevil mortality and progeny production. Ecological farming has globally spread in agricultural practices. The last work phase of this production chain is the control against stored product pests. From this angle, our results can contribute to sustainable agricultural production.

## 2. Materials and Methods

### 2.1. Insect Culture and IR Equipment

*S. granarius* was reared on wheat grains with 13.5% moisture content. Adult insects of both sexes and mixed ages were used for IR irradiation tests. The culture was maintained in optimal conditions for the insects in a climate chamber at 26 ± 2 °C, 60 ± 7% RH and 14/8 photoperiod. All experiments were carried out under the same environmental conditions.

Infrared incandescent (Tungsram 250W 235-245V E27) applied also in animal husbandry were used for our experimental investigations. It has 250W nominal power and 240V nominal lamp voltage. Eight light bulbs covered with aluminum funnels were tied to a wooden pole and hung next to each other in a linear array.

### 2.2. Optimal IR Irradiation Distance Based on the Median Lethal Dose

First, the insect mortality (LD_50_) effects of five different irradiation distances were analyzed, the outcome of which the subsequent crop irradiation experiments were based on. The assessed distances from the bottom of the glass jar to the lamp were 13, 15, 17, 19, and 21 cm. For the experiments seed-free, empty glass jars were used, to which 10 healthy *S. granarius* adults were added. Each treatment included 4 repetitions. The samples were irradiated continuously until 50 percent of the adults used in the experiment perished. The mortality rate was continuously monitored at each distance examined. Four samples per treatment were irradiated at the same time at 28 °C, 60 ± 7% RH. The measured LD_50_ values were represented in a coordinate system as a function of time.

### 2.3. IR Irradiation of the Crop Infested by Insects

Untreated, clean, and infestation-free wheat grains with 13.5% moisture content were used for the experiment. Two different weight groups of wheat grains were formed (50 and 100 g) in advance. Each sample consisting of 20 healthy *S. granarius* adults of mixed sexes and ages was placed in a small glass jar. The glasses were covered with textiles and placed in a climate chamber equipped with a fan at 26 ± 2 °C, 60 ± 7% RH and 14/8 photoperiod. 

To assess the effect of the infrared wave powers on insect-mortality (as compared to the untreated samples), a total of five exposure times of irradiation were set up. The same irradiation power (250 W) was applied continuously for 45, 60, 90, 150, and 210 s (exposure time) and the distance of irradiation was 17 cm (Figure 1). Each treatment included 4 repetitions. Eight samples (4-4 of both of the 50 and the 100 g weight samples of grains) per treatment were irradiated at the same time at 26 ± 2 °C, 60 ± 7% RH. Dead adults were counted after 12, 24, 48, and 72 h.

The adult insects (dead and alive) were removed from each sample after the 72 h-count in order to the cognition of IR irradiation effect on progeny. The jars were returned to the incubator for another 45 d. After this period, the emerged *S. granarius* adults were counted, assessed as dead or alive and removed from the vials. 

### 2.4. Adult-Activity Change by IR Irradiation

The wheat seeds weighing 100 g were placed in small glass jar (same as those employed in the previous experiments) to which 20 healthy *S. granarius* adults of mixed sexes and ages were added. The experiment was repeated four times. The samples were kept in the same circumstances described above. The samples were irradiated by IR lamps (from 17 cm) for 210 s in order to monitor the adult-activity trend triggered by IR irradiation. Snapshots were taken of the samples during the irradiation every 10 s using Canon EOS D80 camera. These records were subsequently evaluated, whereby the degree of activity (DA) was determined. 

DA = m × d where m: is the number of the adults appearing on the wall of the glass; d: is the distance of the adult crawling farthest from the surface of the crop (cm). The trend of these parameterized data was visualized in a diagram used to determine the highest activity period during the irradiation.

### 2.5. Statistical Analysis

Mortality counts were corrected by using Abbott’s [20] formula. In order to test the mortality data of the granary weevil (*n* > 50), the Shapiro–Wilk test was used. For the survey of the distribution of data (*p* < 0.05), the Ghasemi- and Zahediasl-type methods were employed. The data were analyzed using two-way ANOVA in SPSS 11.5 software, with weevil mortality as the response variable as well as exposure time and time elapsed after IR as the main parameters. The mortality values of different weight samples, as well as the numbers of progeny, were also examined statistically by one-way ANOVA. Means were separated by using the Tukey (HSD) test, at *p* ≤ 0.05.

## 3. Results

Median lethal dose (LD_50_) measured at the different distances of infrared irradiation examined can be seen in Figure 2A. Polynomial-type LD_50_ was caused by the linearly increased irradiation distances of the applied infrared power. Mortality curves can be divided into two well-defined parts. The slow decrease of LD_50_ value can be observed in the first half of the tendency until the optimum distance of the irradiation source (17 cm) was reached. The applied power infrared-irradiation proved to be the most effective at this distance. On average, half of the experimental animals were destroyed by the 250 W incandescent lamp located at this distance in 148 s. The tendency of LD_50_—after a slow rise—showed an exponential increase in response to increasing the distance of the source of the infrared irradiation. LD_50_ value triggered by infrared irradiation occurred at the distance of 21 cm in 300 s. It is suggested by this survey that the effectiveness of irradiation primarily depends on the angle of incidence of the emitted electromagnetic waves, which corresponds to the finding of the optimal distance between the radiation source and the surface of the target (Figure 2B).

Insect mortality measured in grain specimen of different weight is shown in Figure 3 as a function of treatments. The Shapiro-Wilk normality test showed that our mortality data are of a normal distribution, *p* > 0.05. Significant differences in the mortality between intact and treated samples were revealed by statistical analysis. The effect of the exposure time of IR irradiation (50 g: df = 4; *p* < 0.001; 100 g: df =4; *p* < 0.001) and the time elapsed after these treatments (50 g: df = 3; *p* < 0.001; 100 g: df = 3; *p* < 0.001) on the triggered mortality data of *S. granarius* were uniformly confirmed by two-way ANOVA for both applied grain weights. The effects of interaction of the two main examined factors (exposure time and the time elapsed after treatment) on the mortality were not statistically significant.

An increase in insect mortality caused by higher exposure time of irradiation and times elapsed after the treatment was evident from 12 h after the onset in both grain weights. The highest exposure time induced the highest mortality in both examined weights: at 210 s and at 50 g grain weight (72 h: 45.45%); and at the same exposure time and at 100 g grain weight (72 h: 64.93%) (Figure 3).

None of the short irradiation periods triggered acceptable mortality/plant protection efficacy. The perishing of insects due to irradiation was evinced in the first time interval that elapsed after the treatment (12 h) in almost every exposure regime; this tendency maintained approximately at the same pace was revealed throughout the longer time intervals measured from the onset of the treatment. This difference in efficacy between each time interval that elapsed after the treatment could not be statistically proven (*p* < 0.05). The complete absence of insect- perishing effects was observed only in one case (at 210 s and at 50 g grain weight in 48 h: 0.00%).

The polynomial-type mortality process (close relationships represented by R squares) was caused by the linearly increased exposure times of irradiation in both grain weights. The insects exposed to progressive time intervals that elapsed following the beginning of the treatment have shown a significant increase in mortality in each exposure time. A significant increase in insect mortality between 150 and 210 s was detected in both weight samples. There were remarkable differences between the mortality rates of samples of different weights. Approximately 1.5–2 times higher mortality values were observed in 100 g weight samples at each exposure time. This difference was corroborated by statistical analysis as well (df = 1; *p* < 0.001). However, a steeper ascending mortality was triggered by the increase of exposure time in higher grain weight (100 g). Independently of these, total mortality of the experimental insects did not occur in any of the applied treatments. According to our calculation based on the acquired polynomial formulas, 100 percent mortality can be expected by 412 s of irradiation at 50 g of grain and at 256 s of irradiation at 100 g weight of grain.

With increasing exposure times, the number of adult progeny decreased at both weights of grain (50 and 100 g) (Table 1). This observation was statistically proven at both experimental weights. After forty-five days, the progeny production of sample populations dropped to approximately 50% at 60 s irradiation times as compared to the intact weevil populations. Adult progeny numbers differed between 50 and 100 g grain weights, but this difference was not statistically proven. Significant progeny decrease was triggered by longer irradiation periods. Nevertheless, a complete suppression of progeny production was not observed, even after the longest exposure time (210 s) applied. Moreover, in both grain weights, the effect of irradiation treatment on the number of dead progeny adults was statistically confirmed uniformly.

In the course of progress of infrared irradiation, the activity curve of adults was characterized by an asymmetric parabolic type (Figure 4A). From the beginning of the irradiation, the activity was continuously increasing, which initially was of a sharply rising character. It fast reached its culminating point, which occurred between the 40–50 s of irradiation. Most adults were located on the wall of the glass jar (Figure 4B) and on the surface of the wheat grains in this period. Subsequently, their activity slowed down and after a longer decay phase the moving of the adults returned to baseline. This decreasing phase lasted for an average of 140 s; three times longer than the introductory period. At the end of the activity, the adults hid among the grains in an attempt to avoid the destructive effects of the irradiation.

## 4. Discussion

Adult-perishing and progeny-suppressive effects of *S. granarius* caused by infrared irradiation were confirmed by our experiment. The efficacy of infrared treatment in plant protection has been bolstered by other studies in connection with a wide range of arthropod pests, including those posing a threat in public health [21,22] and forestry [23], as well as by stored product pests [1,4,8]. From the genus *Sitophilus* primarily, the maize weevil, *S. zeamais* [19,24] and the rice weevil, *S. oryzae* have been examined in previous investigations [19].

According to our results, higher *S. granarius* mortality rates were reached by applying longer exposure times and longer times that elapsed after infrared-irradiation treatments. Additionally, our results show that the setting or the compliance with the suggested distance between the infrared-emitter and the surface to be treated is very important in terms of expected efficacy. It has to be noted that the optimum distance of IR irradiation is influenced by the presence or lack of the infested grain. In fact, the weevils without infested grain had to irradiate for a mean of 148 s to obtain 50% mortality, whereas in the experiments with the grain, more than 21 s were needed to detect 50% mortality, only when dead insects were counted 72 h (50 g) or 48 h (100 g) after the treatment.

Most of former studies were mainly focused on different developmental stages (e.g., egg) [24], the effect of the grain moisture [19] or the impact of the temperature of the grain [18]. According to the results of Khamis et al. [18], the total mortality of *S. oryzae* adults was attained at 8.0 cm from the emitter using 113.5 g of wheat, with a 60 s exposure. Comparing with our results, it is clear that the lower temperature regimes require longer irradiation times to obtain total mortality, because the grain must warm up.

A significant increase in *S. granarius* mortality can be observed in the higher grain weight samples. This higher efficacy can be expounded by the special heating character of infrared as it predominantly heats the opaque, absorbent objects, and higher weight objects can transmit a higher degree of heat [9,10,11]. Nevertheless, this method can be suitable for disinfestation of small volume items, as well as empty warehouse or floors, due to the lack of grain warming preparation.

Previous results pointed out [25] that decreasing seed embryo viability and seedling vigor influence the effect exerted by the energy conveyed by a certain spectrum range of electromagnetic waves. Experimental studies have demonstrated that heat can be produced in seeds during metabolic activities, which can trigger a discernible impairment in response to the changing physiological status of seeds [26]. Therefore, careful selection of irradiation dose and time should be realized while treating seed items. This harmful effect cannot be significant in our case because the model grain was not used as seed, but as forage. From this point of view, the seedling ability is not a determinate character of treated grain in forage utilization.

The activity of pests was increased right after the beginning of irradiation, which was proven by the photo images taken. Naturally, the activity degree of weevils is influenced by several parameters not investigated by us (e.g., glass surface temperature, quality of storage box, air volume, and so on), but the tendency in changes in activity depending on the temperature was reinforced by our results. Combined protection using residual insecticide and some alternative methods have already been known in post-harvest technologies, which were confirmed by several results of pertinent studies [27,28].

Suppressive consequences of the treatment on the progeny have been confirmed by our experimental observations. It is known that infrared irradiation has deleterious effects on insects such as reduction of reproductive rate of *S. oryzae* [29].

Our study pointed out that infrared radiation has insecticidal activity on *S. granarius* adults and a suppressive effect on its progeny. Therefore, infrared radiation may be a promising tool in control strategies as a sustainable means of superseding chemical control in certain situations. Our results may also contribute to developing a more reliable and feasible method for controlling stored product pests.

In summary, properly chosen infrared irradiation (intensity and duration) can be used for the management of *S. granarius* larval and adult developmental stages. This treatment alone, and combined with other solutions, could provide an effective and viable environmental treatment technique in integrated pest management (IPM) programs.

## 5. Conclusions

Infrared radiation may provide a new approach in the development of sustainable protection means against stored product pests, such as weevils (*Sitophilus* spp.). Further investigations could be carried out to examine further the more profound effects of this alternative pest management method. Infrared radiation, especially when combined with other protection methods, can contribute to a successful realization of integrated pest management, on other pestiferous species of stored grains in order to minimize the use of residual insecticides. Naturally, the details of this possible protection method must be improved for rendering it suitable for fulfilling the requirements of the practice, which will be the task of future endeavors exerted in addressing the urgent need for environmentally sounder pest management.

## Figures and Tables

**Figure 1 insects-12-00102-f001:**
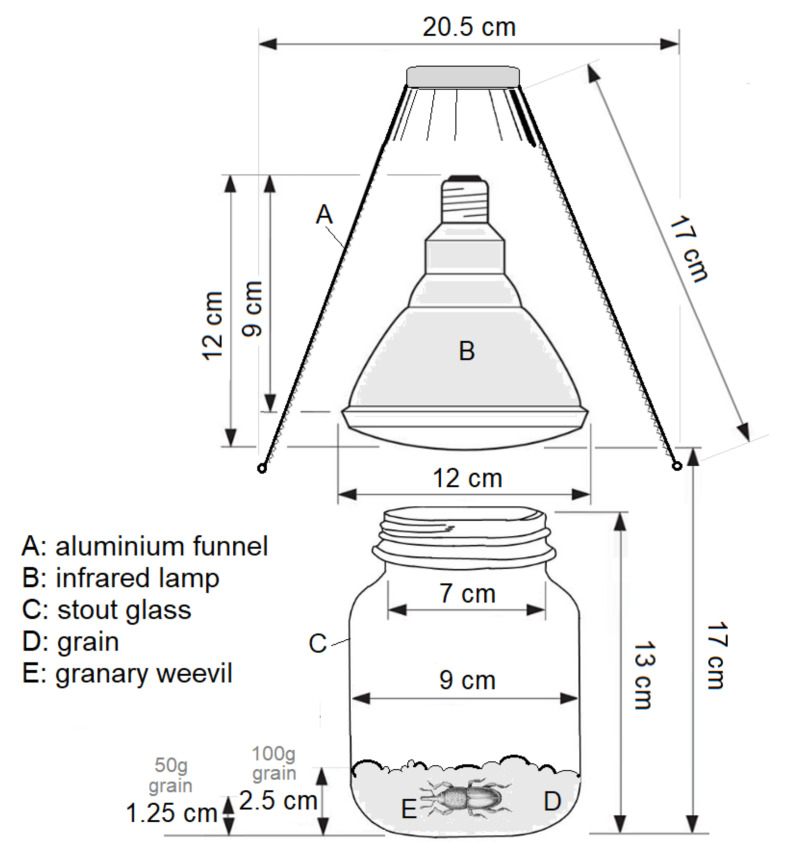
Parameters of the infrared experimental setting.

**Figure 2 insects-12-00102-f002:**
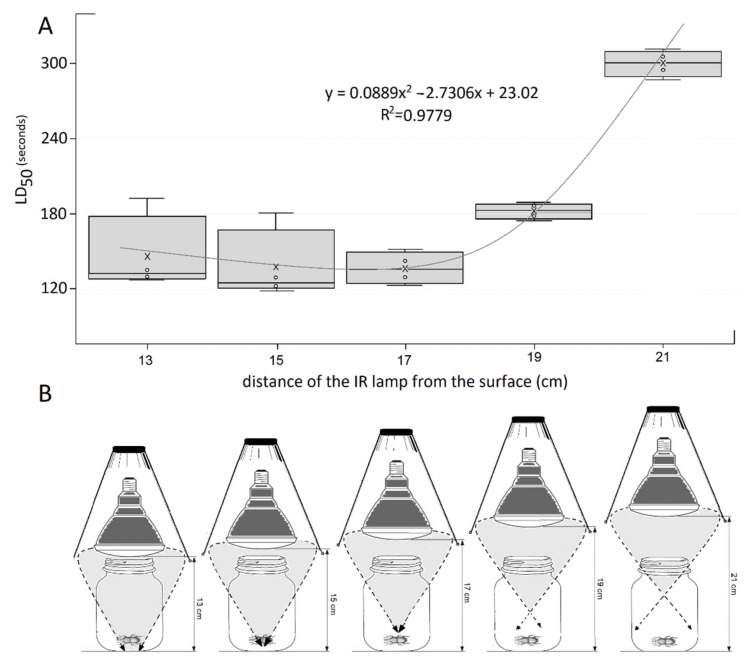
Median lethal dose (LD_50_) at the different distances of infrared irradiation as a function of time. (**A**) Mortality values and other statistical parameters. (**B**) The potential explanation for the efficacy of infrared irradiation (IR) changing depending on the distance from the surface.

**Figure 3 insects-12-00102-f003:**
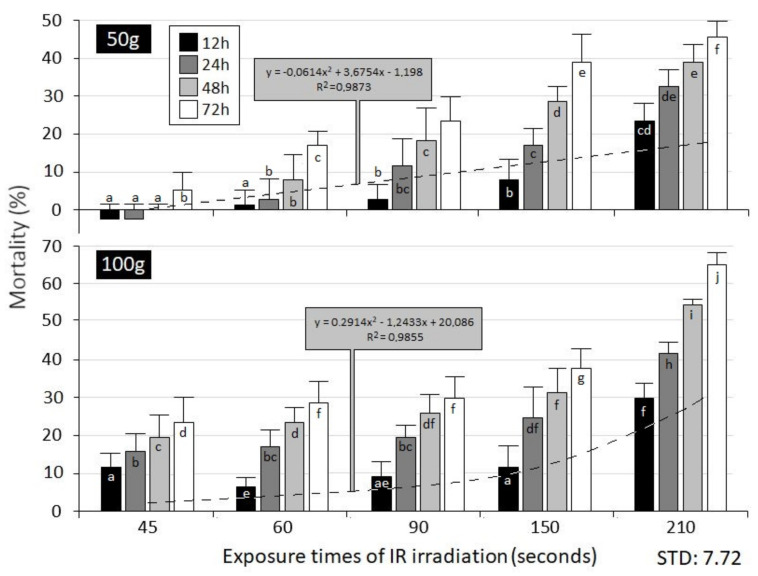
Abbott-corrected percentage of mortality of *S. granarius* adults (mean±SE) treated by IR irradiation, as a function of different exposure times and times elapsed after treatment. Different small letters showed significant differences among treatments according to Tukey’s test.

**Figure 4 insects-12-00102-f004:**
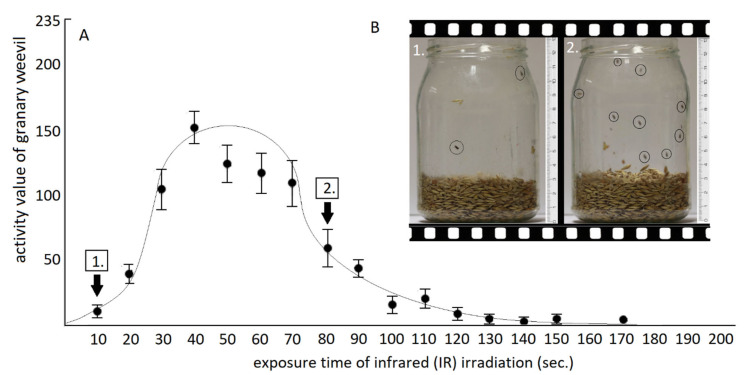
Degree of activity of the granary weevil as a result of the increase in the infrared exposure time. (**A**) Tendency of movement activity; (**B**) Photo images depicting the movement- activity.

**Table 1 insects-12-00102-t001:** Progeny production of *S. granarius* (mean number of adults ± SE) and percentage (% ± SE) of dead progeny on infrared-irradiated wheat, 45 days after the removal of exposed *S. granarius* adults, and the statistical relationships (*p* ≤ 0.05).

		No. Progeny		% Dead Adults		No. Progeny		% Dead Adults	
Weight of Grain		50 g				100 g			
control		21.25 ± 1.10	(a)	11.69 ± 1.61	(a)	37.25 ± 5.57	(a)	9.50 ± 2.63	(a)
exposure times	45 s	14.50 ± 1.19	(b)	25.09 ± 4.08	(b)	15.00 ± 3.08	(b)	34.19 ± 2.56	(b)
	60 s	13.75 ± 4.51	(b)	30.80 ± 9.63	(c)	19.75 ± 1.25	(c)	40.10 ± 2.52	(c)
	90 s	11.25 ± 0.25	(c)	27.66 ± 4.49	(b)	14.00 ± 1.68	(b)	38.37 ± 2.53	(c)
	150 s	7.75 ± 1.25	(d)	32.43 ± 8.90	(c)	14.25 ± 1.88	(b)	37.87 ± 3.35	(c)
	210 s	5.00 ± 1.15	(d)	50.83 ± 6.68	(d)	14.50 ± 0.95	(b)	26.59 ± 2.10	(d)
df = 5; *p* < 0.001									

Different small letters showed significant differences among treatments according to Tukey’s test.

## Data Availability

The mortality data presented in this study are available on request from the corresponding author.

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
