# Peer review of "Effects of Different Infra-Red Irradiations on the Survival of Granary Weevil *Sitophilus granarius*: Bioefficacy and Sustainability"

_insects, 2021, doi:10.3390/insects12020102_

Round 1
Reviewer 1 Report
Dear Authors,
I have now completed my review of "Effects of the Different Infra-Red Irradiations on the Survival of Granary Weevil Sitophilus granarius: Bioefficacy and Sustainability" for Insects, and submitted my conclusion, "Accept after minor revision (corrections to minor methodological errors and text editing)."
My remarks and suggestions for correction and completion can be seen in the PDF document.
Final English control of the text is necessary.
Kind regards,
Reviewer

Author Response
Dear Reviewer 1,
thank you for the consideration of our MS, entitled „ Effects of the Different Infra-Red Irradiations on the Survival of Granary Weevil Sitophilus granarius: Bioefficacy and Sustainability”. We send our revised article based on the opinion of peer-reviewer 1.
We corrected all remarks, mistakes in MS, which are indicated by coloured (red) text using track-changes.
We hope our revised MS will meet the requirements of MDPI Insects.
Thank you very much for your contribution and assistance!
Yours sincerely,
Sándor Keszthelyi

Reviewer 2 Report
This work is not innovative. There is a series of bibliography regarding the use of infrared to treat insect species in stored products, namely Sitophilus spp. The response of the different species of Sitophilus does not seem, in my point of view, to be significantly different.
Usually this method is used to disinfect spaces and not commodities. Although there are published articles that confirm that the germinating power of the seeds is not lost with this treatment. I, honestly, think that a lot of work should be done to confirm, since after 50ºC most living organisms end up dying. And the grain is a living organism.
Another issue that arises here, is the work done in the laboratory with 50gr and 100gr of grain. These results are only indicative since the real conditions are far from these tests.
Finally, S. granarius is not one of the most devastating pest in stored grain in globally. I never caught this species in my country or in Africa.
I advise the authors to explain the added value of this work compared to previously published works
I saw many articles related with this treatment but I can not assure that is plagiarism. I believe not.
Author Response
Dear Reviewer 2,
It gives us pleasure to resubmit our manuscript to MDPI Insects, entitled „ Effects of the Different Infra-Red Irradiations on the Survival of Granary Weevil Sitophilus granarius: Bioefficacy and Sustainability”. We appreciate the reviewer’s affirmation of this study and suggestions on the structure and organization of the writing in the previous manuscript.
We have inserted the asked thoughts into the text and we have fully revised the manuscript according to their comments. We attached the revised MS files via ms center:
- insects-1063253 1MS with track changes Rev2.pdf
- insects-1063253 1MS revised based on Rev2. pdf
Our answers to the remarks and questions of Rev2 are followings:
- The significance of our work was enhanced in our objectives:
- The utilization of IR irradiation in forage crop protection was explained in the discussion.
- Actually, the results of our work are only indicative type, because our aims were to obtain an alternative method of crop protection in ecological practice. Naturally, the details of this possible protection must be improved for the practice, which will be the task of the next scientific researches in days to come.
- The economic importance of the pest and its varying damage values are emphasized at the beginning of the introduction section.
- The added value and predicted benefits were enhanced at the end of the conclusions.
We hope our revised MS will meet the requirements of MDPI Insects.
Thank you very much for your contribution and assistance!
Yours sincerely,
Sándor Keszthelyi

Reviewer 3 Report
Infrared radiation is an efficient and safe physical process method, with wavelengths range from 0.75 to 100 μm. IR can be directly transferred to the material without medium, and converted into heat after the absorption of the electromagnetic wave. Gas-fired IR technology was firstly used for grain disinfestation in 1940s. It is also discovered that the mortality rate increased with the temperature in several insect pests.The sources of infrared radiation in various drying application for stored grains have evolved with time from gas-fired IR to open flame IR to recent flameless catalytic IR. The source temperature at IR emission site is different from grain temperatures. Most studies on control of internal grain feeders using IR revealed a grain temperature range of 60 to 70 ˚C can achieve 100% of mortality. In studies on the potential for flameless catalytic infrared technology at source temperature of 500 ˚C for control of insects such as Rhyzopertha dominica, Sitophilus oryzae and Tribolium castaneum, a significant correlations between mortality and temperatures were found in earlier studies. The impact of IR technology on drying, insect disinfestation and milling quality has been active area of research.
Most of the findings that authors discussed in their manuscript were reported in several other insect species, including rice weevil, Sitophilus oryzae. In the current study authors measured the impact of five different IR irradiation exposure times ranging from 45 s to 210s on adult mortality of S. granarius in wheat seed by using infrared incandescent bulb (250W 235-245V) placed at 17.0 cm above the bottom of glass bottle and did some correlation between exposure time and mortality. High exposure times in this study, presumably produced high temperatures. No measurements on grain temperatures were recorded. For practical applications, information on grain temperature is more relevant. The relationship between insect mortality and temperature of the grain would be different from the relationship between mortality and exposure times.
In another set of experiment , IR irradiation impact on adult-activity was monitored in samples with adult insects irradiated by IR lamps (from 17 cm) for 210 seconds. Movement activity of the granary weevil as a result of the increase in the infrared exposure time was plotted. Increase in activity in weevils within 60 s was manifested by adult presence on the wall of the glass jar and on the surface of the wheat grains. In the later phase of exposure, decreased activity with weevils located under grain, presumably due to high temperatures as medium heats up due to IR. Temperature data from grain could be useful. Glass surface could influence the amount of heat produced by IR irradiation. None of the results in this study were interpreted from practical perspectives.
Number of IR irradiation studies investigated the influence of grain temperature on insect mortalities in several insect species including the related rice weevil (please see the first paragraph). Discussion of results from this study by contrasting with other insect species is needed; however, authors are curtailed to discuss such aspects because of lack of temperature data.
Authors conducted an array of experiments to demonstrate the application of IR irradiation by conducting controlled experiments without essential details of methodology. For instance, the irradiation source used in all these experiments was incandescent bulb (250W 235-245V). Details on irradiance, temperature at the origin are more important in this study.
Author Response
Dear Reviewer 3,
It gives us pleasure to resubmit our manuscript to MDPI Insects, entitled „ Effects of the Different Infra-Red Irradiations on the Survival of Granary Weevil Sitophilus granarius: Bioefficacy and Sustainability”. We appreciate the reviewer’s affirmation of this study and suggestions on the structure and organization of the writing in the previous manuscript.
We have inserted the asked thoughts into the text and we have fully revised the manuscript according to their comments. We attached the files (revised and with tracked changes) via ms center:
Our answers to the remarks and questions of Rev3 are followings:
- However, the earlier studies have primarily deal IR irradiation in warmed grain. In contrast, we wonder about the efficacy of IR irradiation on granary weevil mortality in the stored crops without warming because this method is both useable in practice and economical rentable. Last but not least, this possibility is not caused damage to the treated grain nutrition values.
- Parallel with the previous question the temperature of the treated grain was not measured due to the formerly discussed arguments.
- The circumstances of the irradiation (e.g. glass surface temperature, quality of storage box and air volume, and so on) during the activity analysis was not assessed because the main target of our investigation was to determine the peak and tendency of the adult activity triggered by the IR irradiation. Moreover, this data can be useful for the optimal timing of chemical protection. Naturally, the details of the practical use must be improved by the forthcoming experimentations.
- The asked paragraph was inserted in connection with the results of the former studies dealing with pest disinfestation, which can be seen in the discussion (please see in attached file)
- We would have liked to set up the experimental conditions as close to the natural circumstances as possible. That is why, our starting point was the untreated grain, which is modeled better than the storing conditions.
We hope our revised MS will meet the requirements of MDPI Insects.
Thank you very much for your contribution and assistance!
Yours sincerely,
Sándor Keszthelyi

Round 2
Reviewer 3 Report
However, the earlier studies have primarily deal IR irradiation in warmed grain. In contrast, we wonder about the efficacy of IR irradiation on granary weevil mortality in the stored crops without warming because this method is both useable in practice and economical rentable. Last but not least, this possibility is not caused damage to the treated grain nutrition values.
Ans) Experiments were conducted with a limited amount of grain (50-100 g). How the results from this study are useful for bulk lots when large quantities of grain are treated? Especially considering the depth of the grain and the volume of the grain. IR radiations on crop produces heat in the grain and temperature would vary with the amount of grain.
Parallel with the previous question the temperature of the treated grain was not measured due to the formerly discussed arguments.
Ans) Please see my response to the above rejoinder.
The circumstances of the irradiation (e.g. glass surface temperature, quality of storage box and air volume, and so on) during the activity analysis was not assessed because the main target of our investigation was to determine the peak and tendency of the adult activity triggered by the IR irradiation. Moreover, this data can be useful for the optimal timing of chemical protection. Naturally, the details of the practical use must be improved by the forthcoming experimentations
These findings have little relevance because the conditions of experiments are different from real world scenario.
The asked paragraph was inserted in connection with the results of the former studies dealing with pest disinfestation, which can be seen in the discussion (please see in attached file)
The Previous study conducted by Khamis et al. employed flameless catalytic infrared, and proved that there was significant correlation between mortality and temperatures of the 3 pests according to logistic regression statistics. The conditions IR irradiation by Khamis is different from current study. Khamis et al. used different source of IR, amount of seed, temperature, and other conductions such as distance of IR source may be different.
Author Response
Dear Reviewer,
thank you for the consideration of our MS, entitled „ Effects of the Different Infra-Red Irradiations on the Survival of Granary Weevil Sitophilus granarius: Bioefficacy and Sustainability”.
The language was improved by a native English speaker
Our answers to the remarks are the followings (corrected items are indicated by red text using track-changes).
- However, the earlier studies have primarily deal IR irradiation in warmed grain. In contrast, we wonder about the efficacy of IR irradiation on granary weevil mortality in the stored crops without warming because this method is both useable in practice and economical rentable. Last but not least, this possibility is not caused damage to the treated grain nutrition values.
Parallel with the previous question the temperature of the treated grain was not measured due to the formerly discussed arguments.
Ans) Experiments were conducted with a limited amount of grain (50-100 g). How the results from this study are useful for bulk lots when large quantities of grain are treated? Especially considering the depth of the grain and the volume of the grain. IR radiations on crop produce heat in the grain and temperature would vary with the amount of grain.
Response) Revised as requested. The revision has been inserted into the discussion: Nevertheless, this method can be suitable for disinfestation of the small volume items as well as an empty warehouse or floors due to the lack of grain warming preparation.
- The circumstances of the irradiation (e.g. glass surface temperature, quality of storage box and air volume, and so on) during the activity analysis was not assessed because the main target of our investigation was to determine the peak and tendency of the adult activity triggered by the IR irradiation. Moreover, this data can be useful for the optimal timing of chemical protection. Naturally, the details of the practical use must be improved by the forthcoming experimentations
These findings have little relevance because the conditions of experiments are different from real world scenario.
Response) Our answer was inserted into the discussion: In all likelihood, the activity degree of weevils is influenced by several parameters not investigated by us (e.g. glass surface temperature, quality of storage box and air volume, and so on), but our achieved results can be pointed out the tendency changing of activity caused by temperature increases.
- The asked paragraph was inserted in connection with the results of the former studies dealing with pest disinfestation, which can be seen in the discussion (please see in attached file)
The Previous study conducted by Khamis et al. employed flameless catalytic infrared, and proved that there was significant correlation between mortality and temperatures of the 3 pests according to logistic regression statistics. The conditions IR irradiation by Khamis is different from current study. Khamis et al. used different source of IR, amount of seed, temperature, and other conductions such as distance of IR source may be different.
Response) Unfortunately, we did not find such experimental results, which would have done under completely similar conditions. In our opinion, the results of Khamis et al and our experimental data are comparable, and parallel can be drawn.
We hope our revised MS will meet the requirements of MDPI Insects.
Thank you very much for your contribution and assistance!
Yours sincerely,
Sándor Keszthelyi

Round 3
Reviewer 3 Report
I urge authors to check for grammar, proof read sentences and make necessary changes. In results section, authors need to mention P values according to format of journal. Also mention degrees of freedom.
Introdution
Delete or edit this sentence "Thus, its economical weight significance is very various depending on its distribution area"
Revise the sentence as "application of residual insecticides for management of stored product insects has several negative side-effects"
Revise - "Recently, grain protective methods alternative to insecticides have received increased attention"
Revise- "As insecticidal use becomes more restricted, irradiation is likely to gain increasing attention as an alternative way of grain protection"
Revise "Therefore, electromagnetic irradiation methods such as ionizing-, microwave or infrared radiation that induce stored product-pest sterility and mortality can be considered as viable alternatives"
In fifth paragraph revise first sentence " Infrared technology can be utilized in stored grain pest control, such as in the detection of insect infestation in stored products .." The next sentence is long winded and awkward please consider to revise this "Numerous efforts have already been made in its utilization in connection with its utilization to combat several stored product pests"
Methods 2.2 change the sentence "adults of mixed genders" to "adults of both sexes"
Define what is LD50 in these experiments.
2.3 mention the distance of irradiation source in text also. "To test the effect IR irradiation on progeny, after the 72 hours-count, the adult insects
(dead and alive) from both parts of each sample.." not clear. what is " both parts of each sample"
Methods on "influence of IR irradiation on progeny production" is confusing. Grain with adults were exposed. At 72 h of holding period, following the exposure to IR, dead and live adults were removed from the grain, and observed for progeny production. The last sentence is confusing. These adults were observed for survival and progeny production as an indication measure of reproductive ability". Are these F2 adults?
2.4 delete this "Similarly, to the previous examination"
"Snapshots were taken of the samples during the irradiation every 10 seconds"
how snapshots were taken? Mention in your methods what was used giving details of the equipment used.
Results: P values are infinitesimally small. Change those to according to conventional practice adopted by journal.
"Significant differences (p=0.000) in the
mortality between intact and treated samples were revealed by statistical analysis" -this makes little sense. Why P value is zero?
Look at these P values and check all P values in the manuscript to follow format of journal. Mention degrees of freedom where ever appropriate.
(50g: p=1.68×10−16; 100g: p=4.93×10−16)(50g: p=7.78×10−9; 100g: p=2.60×10−12)
Author Response
Dear Reviewer3,
thank you for the consideration of our MS, entitled „ Effects of the Different Infra-Red Irradiations on the Survival of Granary Weevil Sitophilus granarius: Bioefficacy and Sustainability”.
Our answers to the remarks are the followings (corrected items are indicated by red text using track-changes).
I urge authors to check for grammar, proof read sentences and make necessary changes. In results section, authors need to mention P values according to format of journal. Also mention degrees of freedom.
Introdution
Delete or edit this sentence "Thus, its economical weight significance is very various depending on its distribution area" – revised as requested
Revise the sentence as "application of residual insecticides for management of stored product insects has several negative side-effects"- done
Revise - "Recently, grain protective methods alternative to insecticides have received increased attention" done.
Revise- "As insecticidal use becomes more restricted, irradiation is likely to gain increasing attention as an alternative way of grain protection"- done This sentence have been rephrased previously
Revise "Therefore, electromagnetic irradiation methods such as ionizing-, microwave or infrared radiation that induce stored product-pest sterility and mortality can be considered as viable alternatives" done. This sentence have been rephrased previously
In fifth paragraph revise first sentence " Infrared technology can be utilized in stored grain pest control, such as in the detection of insect infestation in stored products .." The next sentence is long winded and awkward please consider to revise this "Numerous efforts have already been made in its utilization in connection with its utilization to combat several stored product pests" done
Methods 2.2 change the sentence "adults of mixed genders" to "adults of both sexes" done
Define what is LD50 in these experiments. done
2.3 mention the distance of irradiation source in text also. "To test the effect IR irradiation on progeny, after the 72 hours-count, the adult insects
(dead and alive) from both parts of each sample.." not clear. what is " both parts of each sample" done
Methods on "influence of IR irradiation on progeny production" is confusing. Grain with adults were exposed. At 72 h of holding period, following the exposure to IR, dead and live adults were removed from the grain, and observed for progeny production. The last sentence is confusing. These adults were observed for survival and progeny production as an indication measure of reproductive ability". Are these F2 adults? done
2.4 delete this "Similarly, to the previous examination" done
"Snapshots were taken of the samples during the irradiation every 10 seconds"
how snapshots were taken? Mention in your methods what was used giving details of the equipment used. done
Results: P values are infinitesimally small. Change those to according to conventional practice adopted by journal.
"Significant differences (p=0.000) in the
mortality between intact and treated samples were revealed by statistical analysis" -this makes little sense. Why P value is zero? done
Look at these P values and check all P values in the manuscript to follow format of journal. Mention degrees of freedom where ever appropriate. done
We hope our revised MS will meet the requirements of MDPI Insects.
Thank you very much for your contribution and assistance!
Yours sincerely,
Sándor Keszthelyi
